# Impact of Consumer Health Awareness on Dairy Product Purchase Behavior during the COVID-19 Pandemic

**Jiabin Xu †, Jingjing Wang † and Cuixia Li ***

College of Economics and Management, Northeast Agricultural University, Harbin 150030, China; xujiabin0319@neau.edu.cn (J.X.); wangjingjing1115@neau.edu.cn (J.W.)
* Correspondence: licuixia.883@163.com
† These authors contributed equally to this work and should be considered co-first authors.

**Abstract:** Corona Virus Disease 2019 (COVID-19) has led to a reduction in the overall consumption of dairy products in China. How to restore the consumption potential of dairy products and alleviate the serious impact on the dairy market in the post-epidemic period is an urgent problem that needs to be resolved. Based on the survey data of 1780 consumers in 31 provinces (municipalities and autonomous regions) of China, the Heckman two-stage model was used to empirically test the impact of consumer health awareness on dairy product purchase behavior during the COVID-19 pandemic and to further analyze the differences in factors affecting dairy product purchase behavior with the restriction of consumer health awareness. The results showed that the overall level of consumer health awareness after the outbreak of COVID-19 was relatively high. A total of 79% of consumers preferred to buy dairy products after the COVID-19 outbreak, and the proportion of purchased dairy products increased by an average of 17.49%, compared with that before the COVID-19 outbreak. Health change perception, health concern degree, and health habit development in consumer health awareness all have important impacts on the purchase behavior of dairy products. Among them, health change perception and health habit development both positively and significantly affected the purchase intention. Moreover, all three aspects of consumer health awareness positively increased the proportion of dairy product purchases. Difference analysis showed that there were obvious differences among consumer groups with different health awareness in dairy product purchase decisions. Component factor analysis found that, overall, consumer health awareness directly affected the purchase intention and increased the purchase proportion of dairy products. Therefore, policy recommendations are proposed to increase the consumption momentum of dairy products by raising consumer health awareness in the post-epidemic period.

**Keywords:** COVID-19; health awareness; dairy products; purchase behavior; Heckman two-stage model

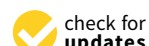



## 1. Introduction

With the continuous promotion of urbanization, the rising income level of residents and the gradual deepening of consumer awareness, dairy products have become an important part of the national economy's "food basket". The consumption potential improvement of dairy products for all people is not only an issue that needs to be focused on after the change of the main social contradiction but also a key task that needs to be overcome for the revitalization of the dairy industry in China. However, the sudden outbreak of COVID-19 in early 2020 quickly hit the pause button on the national economy, and consumption levels and consumer confidence continued to decline. According to survey data from the Tencent Research Institute (from https://baijiahao.baidu.com/s?id=1659755019949382114&wfr=spider&for=pc accessed on 28 February 2020) and China News Network (from http://news.sina.com.cn/o/2020-03-06/doc-iimxxstf6988976.shtml accessed on 6 March 2020), nearly 70% of residents' consumption decreased during the COVID-19 pandemic. Nearly 60% of resident's consumption decreased by more than 20%, and the consumer confidence index in China was 96.57, which fell year-on-year. As the "vegetable basket" product

of the national economy, dairy products have always been a popular one for residents to consume. However, according to data released by the National Bureau of Statistics (from http://www.stats.gov.cn/ accessed on 30 September 2020), since the outbreak of COVID-19, the overall consumption level of dairy products has continued to decline. From January to September 2020, the total profit of dairy product processing was 22,359 billion CNY, with a year-on-year decrease of 13.63%. On 17 August 2020, the number of newly confirmed cases reported in China was zero, which indicates that the domestic epidemic has been largely controlled and indicates that China will enter a post-epidemic period, i.e., a normalized phase of external prevention of importation and internal prevention of rebound (from http://www.nhc.gov.cn/ accessed on 17 August 2020). How to restore dairy consumption potential in the post-epidemic period and mitigate the serious impact of the epidemic on the dairy market are issues that need to be addressed urgently. However, the enhancement of consumption potential needs to start from consumers' subjective awareness, so it would be worthwhile to further investigate whether health awareness can be included in the analytical framework that affects the purchase behavior of consumers of dairy products.

At present, many scholars have conducted extensive studies on the purchase behavior of dairy products, mainly focusing on the influencing factors of purchase behavior. According to the attributes of the influencing factors, they can be divided into the following three dimensions: the first is the dimension of individual characteristics and family endowment characteristics. Individual characteristics can affect the results of purchase behavior [1,2]. Education, income, gender, and age affect the main preferences of consumers for food purchases. Consumers with higher education and income pay more attention to food safety factors, and the elderly and women tend to choose safe food [3]. The second is the dimension of consumption environment and product characteristics. Product packaging, nutritional content, product popularity, product quality, and taste all affect consumer purchase behavior [4]. Product packaging color, packaging design, and innovative ideas can affect consumers' sense of consumption and thus affect the purchase behavior of dairy products [5]. Corporate social responsibility and ethical behavior had positive impacts on the purchase behavior of dairy products [6]. The third dimension is the dimension of consumer trust. Trust in the product's characteristics is crucial for consumer choice [7,8]. Purchase intention is the most used measurement when trying to predict purchase behavior [9,10]; purchase intention based on both expectation and perception are significant in predicting purchase behavior [11].

The study on health awareness originated in the early 1990s, including individual health awareness and healthy lifestyle [12], as well as the relationship between health awareness and healthy behavior [13]. Based on the self-awareness scale, a health awareness scale was developed, and it was pointed out that health awareness was a manifestation of an individual's psychology or inner state. Health awareness refers to the tendency of individuals to pay attention to themselves [14]. Health awareness was a component of health IQ. With health awareness being used to assess the willingness to act on health, health awareness has been gradually introduced into the research field of consumer behavior.

Some scholars have discovered through studies that health issues are the main motivation for buying organic food [15], and health awareness is the key factor that individuals consider. At the same time, some scholars have found that health awareness can better predict consumers' attitudes, intentions, and purchase of organic foods [16,17]. Therefore, it is necessary to study the mechanism of consumer health awareness on purchase behavior. Existing studies have paid attention to the impact of health awareness on the purchase intention of green food [18,19], the purchasing tendency of functional food [20,21], as well as organic food purchase [22], etc. However, these studies only use health awareness as a single explanatory factor that affects purchase behavior instead of focusing on it as a core explanatory variable. The influence mechanism of health awareness on purchase behavior has not yet been discovered, and studies have not yet been found to explore the impact of consumer health awareness on dairy product purchase behavior, especially after the outbreak of COVID-19, consumer health awareness and dairy product purchase

behavior will both change. Analysis on the impact of consumer health awareness on dairy product purchase behavior from both theoretical and empirical perspectives has important theoretical and practical significance.

Previous studies provide a solid foundation for this paper. However, the problems of resuming dairy product purchase behavior need to be further studied during the post-epidemic period: First of all, previous studies have not incorporated the consumer health awareness variable into the impact framework for the analysis of dairy product purchase behavior. In theory, consumer health awareness will inevitably change during the epidemic period, so it is necessary to explore the impact of consumer health awareness on dairy product purchase behavior; Secondly, in previous studies, dairy product purchase was mainly focused on a single research intention or a single research behavior. Although the intention is the dominant decision of behavior, researching only on purchase intention or purchase behavior cannot fully explain the consumption potential of dairy products, so it is necessary to further understand the actual payment level based on the study of purchase intention. In light of this, this paper first analyzed the impact mechanism of consumer health awareness on dairy product purchase behavior at the theoretical level and then combined the survey data of 1780 consumers in 31 provinces (municipalities and autonomous regions) across the country during the post-epidemic period, and used the Heckman two-stage model to empirically examine the impact of consumer health awareness on dairy products purchase behavior; Finally, this paper proposed several policy recommendations based on the research conclusions, so as to provide a decision-making reference for restoring the potential of dairy products consumption and mitigating the impact of the epidemic on the dairy market.

Compared with previous studies, the possible innovations of this paper are as follows:

1.  The uniqueness of the research perspective. The focus of previous studies on dairy product purchase behavior is on the policy environment, market environment, consumer household environment, and other factors, thus ignoring the impact of changes in consumers' own health awareness on consumption behavior. In particular, consumers' health consciousness has fluctuated significantly after the outbreak of COVID-19, and since dairy products have immune-enhancing effects, it is unique to investigate the influence of consumers' health awareness on dairy product purchase behavior.

2.  The systematic nature of the research method. Most of the studies on the purchase behavior of dairy products separate the purchase intention from the purchase behavior. According to the theory of planned behavior, it is known that willingness is the precursor of behavior, and the two studies should be organically combined. This paper adopted the Heckman two-stage model to integrate dairy purchase intention and behavior into the same framework for systematic research, thus avoiding the one-sidedness of using OLS regression and Probit regression.

3.  The timeliness of the research program. A buffer period was given to consumers after the outbreak of COVID-19 so as enhance their awareness of the dangers of COVID-19 and the immune-enhancing effects of dairy products. Then we launched a survey program to observe changes in their health awareness and dairy product purchase behavior. After collecting a valid sample, we adopted a research paradigm that combined theory and empirical evidence and proposed practical policy recommendations based on the findings, and the study itself was time-sensitive.

## 2. Theoretical and Logical Analysis

Health is what human beings want, and the cognition and definition of health are constantly changing over time. Traditionally, health usually refers to the absence of disease, and illness means unhealthy. However, the World Health Organization pointed out in 1948 that health was not a single, clear goal, and it was a complete good state of the body, mind, and society, not just an exemption from disease or weakness. Health awareness is a mental state of individuals' self-recognition of health, which is suitable for assessing the tendency of individuals to adopt healthy behavior, representing the orientation of healthy

lifestyles. Under normal circumstances, individuals with high health awareness pay more attention to their own health and have higher motivations to adopt healthy behavior. At the same time, health awareness can affect the health information process of consumers, and consumers with high health awareness have higher motivations to process relevant information [23]. Studies have shown that health awareness has a significant impact on health-related attitudes and behaviors. In terms of food choices, groups with high health awareness prefer to eat functional foods.

Health awareness is divided into four dimensions, namely, health self-awareness, health alertness, health self-monitoring, and health participation [13]. Based on existing research and the characteristics of the research objectives, this paper divided consumer health awareness into three dimensions: the first dimension is health change perception, which refers to consumers' comparative understanding of their own health perception before and after the outbreak of COVID-19 based on the time dimension, objectively reflecting the perception of physical health changes of consumers [24]; the second dimension is health concern, which refers to whether individual physiological functions caused by external conditions, living environment, personality, and other factors continue to be in good condition, reflecting the degree of consumers' attention to their own health after the outbreak; the third dimension is healthy habit development, which refers to the development of healthy living habits by consumers before and after the outbreak. The dynamic changes reflect whether consumers can maintain healthy living habits for a long time.

It needs to be emphasized that the reason why consumers' health awareness can be linked with dairy product purchase behavior is that dairy products are of great significance to human health. On the one hand, dairy products can provide the human body with rich nutrients; on the other hand, rich protein, active peptides, and other substances in dairy products have significant regulatory effects on humans under the same thermal energy intake condition, thus leading to better body immunity with a more obvious effect.

Therefore, the impact mechanism of consumer health awareness on dairy product purchase behavior can be specifically analyzed from the three dimensions of health awareness: First of all, in terms of health change perception, individual behavior is not only affected by information but also regulated by health cognition according to the protection motivation theory (PMT) analysis. Individuals adopt different coping modes or behaviors according to different cognitions. After the outbreak of COVID-19, the health of consumers was seriously threatened. Consumers with high health awareness understand and know their own health status and changes so that they may be more proactive in maintaining and improving their own health status. The simplest and most important method is to improve human immunity through dietary improvement so that people may be more willing to buy dairy products. Secondly, in terms of health concern, individuals with a higher degree of health concern are more sensitive to changes in the external environment according to the idea of continuous attention theory (CFT). Faced with the impact of COVID-19, the self-protection awareness of people is constantly increasing, and the degree of attention to their own health is also significantly increased. Therefore, the motivation of consumers to buy dairy products may be out of consideration for the health of themselves and their families. That is, the higher the health concern of consumers is, the greater the purchasing power of dairy products is. Finally, in terms of habit development, reinforcements are necessary to the establishment and formation of any new behavior according to the theory of habit formation (HFT), and repetitive actions of reinforcements would promote the generation, development, improvement, and consolidation of new behavior. COVID-19 is a public health emergency. Although the country has used the shortest time to suppress its spread, the normalization of epidemic prevention and control will be a long process, and this long process will also promote the health habits of consumers. The development of dairy products will increase purchase intention of dairy products and further affect the purchase behavior. The theoretical framework of this paper is constructed through the above theories, as shown in Figure 1.

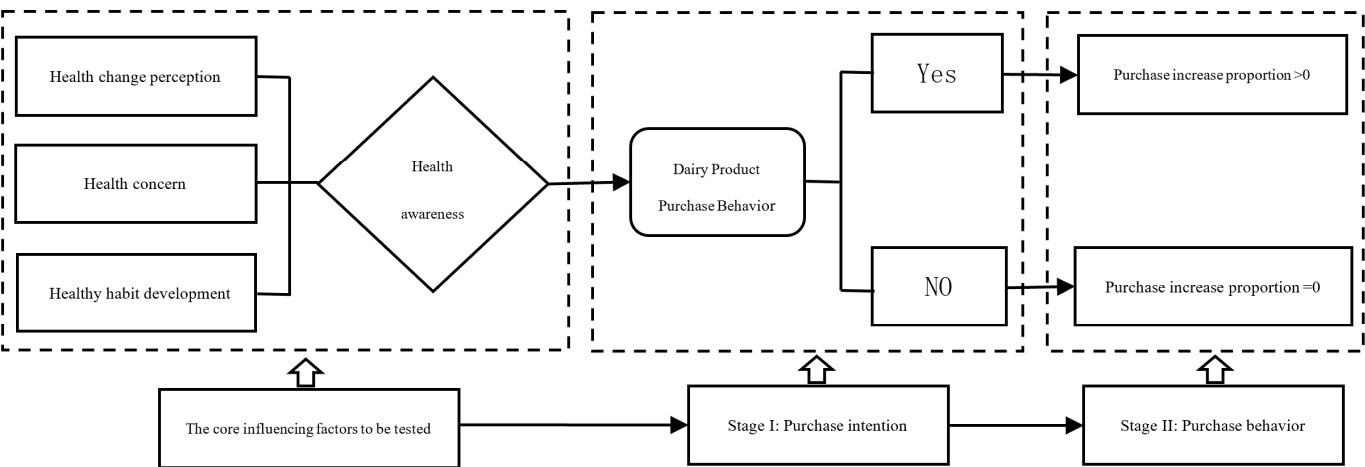

**Figure 1.** Theoretical Framework.

### 3. Research Design

*3.1. Data Sources and Sample Characteristics*

Since the outbreak of COVID-19, the Animal Husbandry Economics Innovation Team of Northeastern Agricultural University has continued to monitor changes in the Chinese dairy market and analyze the direction of dairy market development. Taking into account the actual characteristics of the development of the epidemic in China, the group took 17 August 2020 as the time point to achieve zero newly confirmed cases and made the subsequent phase a post-epidemic period, i.e., a normalized phase of external prevention of importation and internal prevention of rebound. From 1 to 20 September 2020, the team designed a questionnaire around the selected topic of this study and conducted an offline pre-survey from 21 to 30 September to correct any unreasonable points in the questionnaire. From 1 October to 30 November 2020, members of the research team distributed questionnaires online through the Questionnaire Star terminal platform. In order to ensure the scientific and comprehensive sample selection, the research sites cover 31 provinces (municipalities and autonomous regions) (Hong Kong, Macau, and Taiwan are not included), and the selection of survey respondents is not limited to gender, age, education level, and others, based on the principle of random sampling of consumer groups in the operation of the market economy. From 1 to 15 December 2020, members of the subject team counted and analyzed the survey data, eliminated and revised questionnaires that were often shorter than 300 s to fill out and those that were illogical. A total of 1876 questionnaires were collected, and there were 1780 valid questionnaires after data verification, with an efficiency rate of 94.88%. The samples were evenly distributed from 31 provinces (municipalities and autonomous regions) (as shown in Table 1), fully reflecting the comprehensiveness of the data. Therefore, this paper will analyze the impact of consumer health awareness on dairy product purchase behavior based on a sample of 1780 people during the impact of the COVID-19. It is important to emphasize that since this paper uses survey data during the post-epidemic period, the questionnaire was designed to ensure as much as possible that the changes in consumer health awareness were caused by the impact of COVID-19 by adding the prominent words "what I would do after the epidemic" to ensure as much as possible that the changes in consumer health awareness. The questionnaire was designed to ensure that the changes in consumer health awareness and dairy purchase behavior were due to the impact of COVID-19.

As for the basic characteristics of the surveyed consumers, there were 485 males and 1295 females, accounting for 27.25% and 72.75%, respectively. The age was mainly concentrated in 18–26 years old, accounting for 80.11%. Most of the samples are students, accounting for 67.75%. The education level is mainly concentrated in college education and above, accounting for 90.67%. On the one hand, it reflects the limitations of online electronic surveys; paper questionnaire surveys found that women, students, and younger consumers

are the key consumer groups for all types of dairy products. Meanwhile, according to the basic characteristics of surveyed consumer households, 55.34% of consumers have elderly or children, 79.89% of consumers have settled in cities and towns for a long time, and 20.11% of consumers have settled in rural areas for a long time, and families with three persons, account for 47.98%. Consumer household disposable income may have a certain impact on the purchase behavior of dairy products. For this reason, during the investigation process, we further observe the changes in consumer household monthly disposable income after the COVID-19 outbreak. We found that 70.56% of household monthly disposable income was basically unchanged, but 12.42% of household monthly disposable income decreased by more than 1000 CNY, and 11.80% decreased by 0–1000 CNY. However, there was also some households' monthly disposable income that increased, with 3.03% increased by 0–1000 CNY and 2.19% increased by more than 1000 CNY (Table 2).

Dairy products are generally divided into seven categories, which include liquid milk, milk powder, condensed milk, milk fat, cheese, milk ice cream, and other dairy products. Liquid milk includes sterilized milk and yogurt. Milk powder includes whole milk powder, skimmed milk powder, whole sweetened milk powder, flavored milk powder, infant milk powder, and other formula milk powders. Condensed milk includes full-fat sugar-free condensed milk (evaporated milk), full-fat sweetened condensed milk, flavored condensed milk, formula condensed milk. Milk fat includes cream for cakes and cream commonly eaten with bread. Cheese includes raw cheese and processed cheese. Milk ice cream includes milk ice cream, milk ice and others, and other dairy products include casein, lactose, and milk tablets. From a survey on the characteristics of 1780 consumer purchase categories of dairy products, most consumers still mainly buy liquid milk. A total of 92.42% of consumers have bought liquid milk dairy products, followed by milk ice cream and milk powder. Condensed milk, milk fat, cheese, and other dairy products are currently consumed in low proportions, with percentages of 28.76% and 28.09%, respectively. The current consumption ratio of condensed milk, milk fat, cheese, and other dairy products is not high, with percentages of 15.11%, 12.13%, 11.18%, and 7.58%, respectively (Table 3).

**Table 1.** Regional Distribution and Proportion of Survey Samples.

| Province | Sample Number | Proportion (%) | Province | Sample Number | Proportion (%) | Province | Sample Number | Proportion (%) |
|---|---|---|---|---|---|---|---|---|
| Beijing | 105 | 5.90 | Anhui | 48 | 2.70 | Sichuan | 50 | 2.81 |
| Tianjin | 76 | 4.27 | Fujian | 49 | 2.75 | Guizhou | 31 | 1.74 |
| Hebei | 45 | 2.53 | Jiangxi | 43 | 2.42 | Yunnan | 35 | 1.97 |
| Shanxi | 34 | 1.91 | Shandong | 66 | 3.71 | Tibet | 45 | 2.53 |
| Inner Mongolia | 72 | 4.04 | Heilongjiang | 137 | 7.70 | Shaanxi | 48 | 2.70 |
| Liaoning | 57 | 3.20 | Hubei | 107 | 6.01 | Gansu | 39 | 2.19 |
| Jilin | 67 | 3.76 | Hunan | 52 | 2.92 | Qinghai | 50 | 2.81 |
| Henan | 82 | 4.61 | Guangdong | 37 | 2.08 | Ningxia | 47 | 2.64 |
| Shanghai | 91 | 5.11 | Guangxi | 36 | 2.02 | Xinjiang | 38 | 2.13 |
| Jiangsu | 34 | 1.91 | Hainan | 43 | 2.42 | | | |
| Zhejiang | 57 | 3.20 | Chongqing | 59 | 3.31 | | | |

*3.2. Model Construction*

Consumer purchase behavior is usually a purchase decision-making process implemented under the drive of intention. It can be divided into two stages, in which the first stage analyzes their intention and the second stage analyzes their behavior. Therefore, taking COVID-19 as the background, this paper explored the effect of consumer health awareness on the purchase intention of dairy products as the starting point. This paper takes "whether consumers are more willing to buy dairy products after the outbreak" as the dependent variable in the first stage and takes "how much the actual purchase of dairy products by consumers has increased after the outbreak" as the dependent variable in the

second stage. For solving such a two-stage problem, the Heckman two-stage model [25] can be used to effectively reduce the influence of sample selectivity bias. This model can test and correct the possible selectivity bias of the sample. The model building process is as follows:

**Table 2.** Basic Characteristics of Survey Samples.

| Statistical Categories | Class Indication | Sample Number | Proportion (%) | Statistical Categories | Class Indication | Sample Number | Proportion (%) |
|---|---|---|---|---|---|---|---|
| Gender | Males | 485 | 27.25 | | Student | 1206 | 67.75 |
| | Female | 1295 | 72.75 | | Civil Servant | 162 | 9.10 |
| Ages | <18 | 27 | 1.52 | Job | Employee | 223 | 12.53 |
| | 18–26 | 1426 | 80.11 | | Individual dealers | 47 | 2.64 |
| | 27–35 | 168 | 9.44 | | Others | 142 | 7.98 |
| | 36–44 | 43 | 2.42 | | 1 | 36 | 2.02 |
| | 45 and above | 116 | 6.52 | | 2 | 98 | 5.51 |
| Education level | Primary school diploma and below | 35 | 1.97 | Family Population | 3 | 854 | 47.98 |
| | Junior high school diploma | 69 | 3.88 | | 4 | 496 | 27.87 |
| | Senior high school diploma | 62 | 3.48 | | 5 and above | 296 | 16.63 |
| | Bachelor | 1025 | 57.58 | | Decrease over 1000 | 221 | 12.42 |
| | Postgraduate and above | 589 | 33.09 | Changes in monthly disposable income after the outbreak | Decrease 0–1000 | 210 | 11.80 |
| Is there an elderly or a child | Yes | 795 | 44.66 | | constant | 1256 | 70.56 |
| | No | 985 | 55.34 | | Increase 0–1000 | 54 | 3.03 |
| Settlements | Urban | 1422 | 79.89 | | Increase over 1000 | 39 | 2.19 |
| | Rural | 358 | 20.11 | | | | |

**Table 3.** Consumer Category Characteristics of Dairy Products Purchase.

| Category | Sample Number | Proportion (%) |
|---|---|---|
| liquid milk | 1645 | 92.42% |
| milk powder | 500 | 28.09% |
| condensed milk | 269 | 15.11% |
| milk fat | 199 | 11.18% |
| cheese | 135 | 7.58% |
| milk ice cream | 512 | 28.76% |
| other dairy products | 216 | 12.13% |

Stage I, construct the Probit model to estimate the binary model to observe whether you are more willing to buy dairy products after the outbreak of the epidemic, where the decision equation can be:

$$Z_i = \begin{cases} 1, & if \quad Z_i^* \geq 0 \\ 0, & if \quad Z_i^* \leq 0 \end{cases} \tag{1}$$

$$Z_i^* = \beta_{1i} X_{1i} + \varepsilon_{1i} \tag{2}$$

In Equations (1) and (2), $Z_i^*$ represents the probability of the decision-making behavior, denoting whether consumers are willing to buy dairy products after the outbreak. $X_{1i}$

represents an effect factor indicating whether consumers are more willing to buy dairy products, containing core explanatory variables; health awareness and control variables; individual consumer characteristics, family endowment characteristics, market environment characteristics, etc. $\beta_{1i}$ represents the estimated coefficients of the corresponding variables. $\varepsilon_{1i}$ is the random perturbation term and $\varepsilon_{1i} \sim N(0, \sigma^2)$.

The formula for calculating the inverse Mills ratio from the estimation results of the Probit model is:

$$\lambda = \frac{\varphi(X_{1i}\beta_i/\sigma_0)}{\Phi(X_{1i}\beta_i/\sigma_0)} \tag{3}$$

In Equation (3), $\Phi(X_{1i}\beta_i/\sigma_0)$ represents the cumulative distribution function, and $\varphi(X_{1i}\beta_i/\sigma_0)$ represents the density function of the standard normal distribution.

In stage II, we select the samples with $Z_i = 1$ and estimate with OLS. According to Probit estimation equation $P(Z_i = 1 \mid X_{1i}) = \Phi(X_{1i}\beta_i)$, we can obtain the estimated value $\hat{\lambda}$, which can correct the sample selection bias as an additional variable in the equation. Introducing $\hat{\lambda}$ to the equation that consumers actually increase the proportion of purchased dairy products after the outbreak, we can construct the OLS regression equation as:

$$Y_i = \beta_{2i}X_{2i} + \alpha\hat{\lambda} + \varepsilon_{2i} \tag{4}$$

In Equation (4), $Y_i$ represents the increased proportion of purchased dairy products after the outbreak, which can be observed only when $Z_i = 1$. $X_{2i}$ represents the factor influencing consumers to actually increase the proportion of purchased dairy products, containing core explanatory variable health awareness, as well as control variables, individual consumer characteristics, family endowment characteristics, market environment characteristics, etc. $\beta_{2i}$ represents the coefficients of the corresponding variables. $\alpha$ is the estimation coefficient of the inverse Mills ratio. $\varepsilon_{2i}$ is the random perturbation term and $\varepsilon_{2i} \sim N(0, \sigma^2)$ with the correlation coefficient between $\varepsilon_{1i}$ and $\varepsilon_{2i}$ as $\text{cor}(\varepsilon_{1i}, \varepsilon_{2i}) = \rho$.

It is worth noting that the Heckman two-stage model requires $X_{2i}$ to be the strict subset of $X_{1i}$, which means that any explanatory variable in formula (4) should also be an explanatory variable in formula (3), and at least one explanatory variable in (3) does not exist in (4). Hence, there should be at least one explanatory variable, which only affects whether consumers are willing to buy more dairy products but does not affect the proportion of more purchased dairy products. As a result, this paper introduces the convenience of consumption into the model. This paper argues that consumer convenience affects whether consumers are more willing to buy dairy products after an outbreak but does not affect how much consumers actually buy more dairy products after an outbreak. The reason is that since consumer convenience refers to the convenience of purchasing dairy products, if consumers do not have convenient access to dairy products, such as the point of sale is far from their place of residence, or delivery is not possible, it will affect consumers' willingness to purchase dairy products, but it does not affect the proportion of dairy products purchased. Therefore, the consumption convenience variable will appear in the Probit regression model and not in the OLS regression model.

*3.3. Variable Selection and Description*

1. Dependent variables. The dependent variables in this paper include purchase intention and purchase behavior. In order to fully investigate the purchase behavior of consumers for dairy products since the outbreak of COVID-19, this paper used consumer purchase intention of dairy products and the actual purchase level as measures of their purchase behavior. The specific survey question is "Is it more willing to buy dairy products after the outbreak?" If the answer is "Yes", you can continue to ask, "How much has the actual purchase of dairy products increased after the outbreak?". The first question corresponds to the dependent variable that reflects the purchase intention, and the second question corresponds to the dependent variable is

the proportion of preferring purchase more, that is, consumer purchase behavior of dairy products.

2. Core independent variables. This paper divided consumer health awareness into three dimensions, namely, health change perception, health concern degree, and health habit development, which were used as core independent variables. The development and measurement of the scale draw on the research of Gould [13]. Concerning the health change perception, consumers are asked, "I will be more alert to whether the body is unwell after the epidemic occurs" and "I will take the blood pressure and clearly know my blood pressure after the epidemic occurs" and "After the outbreak, I will take regular physical examinations to understand the changes in my health". With regard to the health concern degree, consumers are asked, "I will pay more attention to my health after the outbreak" and "I will take the initiative to read the relevant information about health promotion after the outbreak" and "I will often exchange general knowledge about health with others after the outbreak". As for health habit development, consumers are asked about "I will stick to three meals a day and maintain a balanced nutritional intake after the outbreak", "I will continue to engage in recreational physical activities after the outbreak", and "I will exercise at least 3 times a week after the outbreak, each lasting 20–30 min", as well as "After the outbreak, I will guarantee to sleep within 6–8 h every day. Consumers choose 1 to 5 points for the above questions based on their actual situation, where 1 to 5 represents "strongly disagree", "disagree", "general", "agree", and "strongly agree", respectively. The higher the score, the higher the level of consumer health awareness level.

3. Control variables. According to previous studies, individual characteristics of consumers (gender, ages, education levels), family characteristics (number of family population, family income change, old people or children, and the place of residence), and consumer trust (quality and safety attention, satisfaction with quality supervision) are included as control variables into the regression equation. Among them, changes in household income are measured by considering the "changes in household monthly disposable income after the outbreak of the epidemic". The options are "reduction of more than 1000 CNY", "reduction of 0 to 1000 CNY", "basically unchanged", and "increase of 0–1000 CNY" and "an increase of 1000 CNY or more" are assigned values of 1 to 5, respectively; the settlement place is measured by living in "rural" or "urban", and the quality and safety concerns in consumer trust are measured by the question "what are your concerns about the quality of dairy products?" and "what is the degree of concern about security incidents? ". The options are assigned in turn from "never concern", "basically not concerned", "general", "always concerned", and "always concerned" in order of 1 to 5, respectively. The quality supervision satisfaction is measured by the question "what is your satisfaction with the current dairy product quality supervision?" to measure, the options are assigned from "very dissatisfied", "dissatisfied", "fair", "satisfied", and "very satisfied" in order of 1 to 5.

4. Identify variables. Considering that convenience of consumption will affect purchase intention for dairy products rather than purchase behavior, this paper chose the convenience of consumption as the identification variable of the Heckman two-stage model. Consumption convenience is measured by the question "Is it convenient for you to buy dairy products?", and the options are assigned values from 1 to 5 representing "very inconvenient", "inconvenient", "general", "convenient", and "very convenient", respectively. The meaning of specific variables and descriptive statistics are shown in Table 4.

**Table 4.** Variable Meaning and Descriptive Statistics.

| Type | Variable | Code | Content | Mean | S.D. |
|---|---|---|---|---|---|
| Dependent variables | Purchase intention | $Y_1$ | Is it more willing to buy dairy products after the outbreak | 0.79 | 0.40 |
| | Purchase behavior | $Y_2$ | How much has the actual purchase of dairy products increased after the outbreak? | 17.39 | 17.49 |
| Core independent variables | Health change perception | $Z_1$ | I will be more alert to whether the body is unwell after the epidemic occurs | 3.91 | 1.10 |
| | | | I will take the blood pressure and clearly know my blood pressure after the epidemic occurs | 2.92 | 1.18 |
| | | | After the outbreak, I will take regular physical examinations to understand the changes in my health | 3.20 | 1.11 |
| | Health concern degree | $Z_2$ | Regarding the health concern degree, consumers are asked | 4.19 | 0.81 |
| | | | I will take the initiative to read the relevant information about health promotion after the outbreak | 4.00 | 0.88 |
| | | | I will often exchange general knowledge about health with others after the outbreak | 3.81 | 0.94 |
| | Health habit development | $Z_3$ | I will stick to three meals a day and maintain a balanced nutritional intake after the outbreak | 3.89 | 0.92 |
| | | | I will continue to engage in recreational physical activities after the outbreak | 3.74 | 0.92 |
| | | | I will exercise at least 3 times a week after the outbreak, each lasting 20–30 min | 3.39 | 1.09 |
| | | | After the outbreak, I will guarantee to sleep within 6–8 h every day | 3.86 | 0.95 |
| Control variables | Gender | $X_1$ | 0 = female; 1 = males | 0.27 | 0.83 |
| | Age | $X_2$ | 1 < 18; 2 = 18–26; 3 = 27–35; 4 = 36–44; 5 = 45 and above | 2.32 | 0.83 |
| | Education levels | $X_3$ | 1 = primary school and below. 2 = junior high school. 3 = High school/secondary school. 4 = undergraduate/college. 5 = Postgraduate and above | 4.16 | 0.82 |
| | Number of family population | $X_4$ | Actual value of family population | 3.52 | 0.90 |
| | Elder people or children | $X_5$ | 0 = No; 1 = Yes | 0.55 | 0.50 |
| | Family income change | $X_6$ | "reduction of more than 1000 CNY", "reduction of 0 to 1000 CNY", "basically unchanged", and "increase of 0–1000 CNY" and "an increase of 1000 CNY or more" are assigned values of 1 to 5, respectively | 2.71 | 0.80 |
| | The place of residence | $X_7$ | 0 = rural; 1 = urban | 0.80 | 0.40 |

**Table 4.** *Cont.*

| Type | Variable | Code | Content | Mean | S.D. |
|---|---|---|---|---|---|
| | Quality and safety attention | $X_8$ | what are your concerns about the quality of dairy products? | 3.45 | 0.86 |
| | Satisfaction with quality supervision | $X_9$ | what is the degree of concern about security incidents? | 3.29 | 0.89 |
| Identify variables | Consumption convenience | $W_1$ | Is it convenient for you to buy dairy products? | 3.69 | 1.09 |

## 4. Results and Discussion

### 4.1. Analysis of Expected Statistical Results

On the whole, 79% of the surveyed 1780 consumers are more willing to buy dairy products after the COVID-19 outbreak, and the proportion of purchased dairy products is higher than that before the outbreak, with an average increase of 17.49%. In order to further test the impact of consumer health awareness on the dairy product purchase behavior under the impact of COVID-19, this paper weights and averages the specific indicators under the three dimensions, such as health change perception, health concern degree, and health habit development. To facilitate the expected statistical analysis and empirical analysis, the weighted average value is further processed by assigning a value of 1 to 3. Specifically, the value is assigned to "1 = smaller" when the weighted average of each dimension index is between 1 and 2 (inclusive), and the value is assigned to "2 = normal ", when each dimension index is between 2 and 4 (exclusive), and the value is assigned to "3 = larger", when each dimension index is between 4 (inclusive) and 5. From the cross-statistical results of independent variables and dependent variables (as shown in Table 4), consumer purchase intention and purchase behavior show certain differences between groups after grouping. From the perspective of the overall changes in consumer health awareness, 66.67% of those with less health concern degree are more willing to buy dairy products, with an average increase in the purchase proportion of 8.93%, and 90.72% of those with greater health concern degree are more willing to buy dairy products with the increased proportion by 27.46%; 61.54% of those who are less concerned about health are more willing to buy dairy products. The average increase in the purchase proportion is 3.13%. A total of 85.33% of those who are more concerned about health are more willing to buy dairy products. The average increase in purchase proportion is 21.43%. A total of 56.46% of those with less healthy habits are more willing to buy dairy products, with an average increase in purchase proportion of 8.47%, 87.89% with better healthy habits are more willing to purchase dairy products, with an average increase in purchase proportion of 22.81%. Although there are differences in the linear ratios between the changes in various dimensions of consumer health awareness and the changes in purchase intention and purchase behavior, to a certain extent, it can be initially found that higher health change perception, health concern degree, and health habit development leads to stronger purchase intention and higher increase of purchase proportion (Table 5).

### 4.2. Analysis of Empirical Results

This paper uses Stata 14.0 statistical analysis software to estimate the Heckman two-stage model and empirically examines the impacts of consumer health awareness on dairy products purchase behavior under the impact of COVID-19. Model (1) is a benchmark model regression, which mainly incorporates individual characteristics of consumers, family endowments, and market environment characteristics into the model for benchmark testing. Taking the impact of health awareness on dairy product purchase behavior into account, three sets of core explanatory variables, health change perception, health concern degree, and health habit development, are included in Model (2). The specific regression results are shown in Table 6. According to the regression results, it can be found that the inverse Mills ratio has passed the significance test in both Model (1) and Model (2),

indicating a sample selection bias. The Heckman two-stage model can effectively solve this problem when this paper focuses on the impact of consumer health awareness on dairy product purchase behavior. Results analysis can be conducted based on Model (2).

**Table 5.** Cross Analysis of independent and dependent Variables.

| Variable | Value | Purchase Intention (%) | | Purchase Behavior (%) |
|---|---|---|---|---|
| | | **Willing** | **Unwilling** | |
| Health change perception | small | 66.67 | 33.33 | 8.93 |
| | normal | 76.73 | 23.27 | 13.56 |
| | large | 90.72 | 9.28 | 27.46 |
| Health concern degree | less | 61.54 | 38.46 | 3.13 |
| | normal | 71.45 | 28.55 | 10.77 |
| | more | 85.33 | 14.67 | 21.43 |
| Health habit development | small | 56.46 | 41.54 | 8.47 |
| | normal | 74.14 | 25.86 | 12.77 |
| | large | 87.89 | 12.11 | 22.81 |

For the impact of consumer health awareness on dairy product purchase behavior, health change perception has passed the significance test at the level of 1% in both the first and second stages, and the coefficient symbol is positive, which means the stronger health change perception of consumers is, the stronger the willingness to buy dairy products, and the higher the proportion of purchased dairy products is. It can be explained as follows: after the COVID-19 outbreak, more consumers can perceive changes of their health conditions, and they will pay more attention to their own nutrition. The balance of dairy products will increase nutrient intake accordingly, thus leading to the increase of purchase intention and purchase proportion. Health concern degree passes the significance test at the 1% level in the second stage, and the sign of the coefficient is positive. That is, the more consumers that pay attention to their health, the proportion of dairy products purchased would increase accordingly. This can be explained by the fact that COVID-19 has a huge impact on consumers' psychology. More people realize the importance of disease prevention and physical health, and they will continue to pay more attention to health to improve their immunity through the ingestion of nutrients. Although dairy products have no medicinal function, their benefits to human health have been proven. Therefore, the increased attention of consumers will help increase the proportion of dairy products they buy. Health habit development passes the significance test at the 1% level in both the first and second stages, and the coefficient symbol is positive. That is, the higher degree of health habit development is, the stronger the purchase intention is. Meanwhile, the proportion of purchased dairy products can also increase. This observation can be explained as follows: dairy products, as a "vegetable basket" product of the national economy, are a standing consumer product in most households. Many households have the habit of drinking liquid milk and eating dry dairy products for a long time. A step-by-step improvement of health habit development will strengthen the purchase intention and increase the purchase proportion. In summary, consumer health awareness has a significantly positive impact on dairy product purchase behavior.

**Table 6.** Regression Results of the Heckman Two-stage Model.

| Variable | Model (1) | | Model (2) | |
| --- | --- | --- | --- | --- |
| | Purchase Intention (Probit) | Purchase Behavior (OLS) | Purchase Intention (Probit) | Purchase Behavior (OLS) |
| Health change perception | | | 0.2031 *** (0.0774) | 0.0755 *** (0.0088) |
| Health concern | | | 0.0426 (0.0852) | 0.0436 *** (0.0104) |
| Health habit formation | | | 0.2960 *** (0.0825) | 0.0309 *** (0.0097) |
| Gender | 0.1780 ** (0.0852) | 0.0211 *** (0.0106) | 0.1344 (0.0871) | 0.0105 (0.0096) |
| Age | 0.0223 (0.0488) | 0.0008 (0.0058) | −0.0217 (0.0507) | −0.0113 ** (0.0053) |
| Education level | 0.0956 ** (0.0426) | 0.0055 (0.0065) | 0.0830 * (0.0430) | −0.0014 (0.0060) |
| Family population | −0.0949 ** (0.0466) | 0.0041 (0.0061) | −0.1149 ** (0.0474) | −0.0047 (0.0056) |
| Is there an old man or a child | 0.3725 *** (0.0833) | 0.0025 (0.0111) | 0.3963 *** (0.0854) | 0.0062 (0.0101) |
| Changes in household income | 0.1102 ** (0.0440) | −0.0075 (0.0065) | 0.1209 *** (0.0448) | −0.0019 (0.0059) |
| Settlements | −0.2259 ** (0.0962) | −0.0068 (0.0125) | −0.1687 * (0.0979) | 0.0017 (0.0113) |
| Attention to quality and safety | 0.0452 (0.0529) | −0.0358 *** (0.0079) | 0.0284 (0.0541) | −0.0299 *** (0.0072) |
| Quality Supervision Satisfaction | 0.2105 *** (0.0500) | 0.0337 *** (0.0079) | 0.1940 *** (0.0514) | 0.0208 *** (0.0072) |
| Consumption convenience | 0.4324 *** (0.0344) | | 0.4118 *** (0.0352) | |
| Con. | −1.9629 (0.3342) | 0.1887 *** (0.0571) | −2.8524 *** (0.3699) | −0.0883 (0.0618) |
| Inverse Mills ratio | −0.0788 *** (0.0289) | | −0.0462 * (0.0277) | |
| Wald chi2 | 35.29 | | 225.07 | |
| Pro > chi2 | 0.0001 | | 0.0000 | |

Note: The parentheses are the standard errors corresponding to the estimated coefficients. ***, **, and * indicate significance at the levels of 1%, 5%, and 10%, respectively. Wald chi2 refers to the significance of the overall parameters of the model, and Pro > chi2 refers to the *p*-value of the significance level of the Wald test.

As for other factors that affect consumer dairy product purchase behavior, age negatively affects dairy product purchase behavior. It can be explained by the fact that older consumers have lower drinking and eating preferences for dairy products than younger consumers. The increase in the purchase proportion would also be lower than that of younger consumers. The level of education positively affects the purchase willingness significantly. The higher the education level of consumers is, the stronger awareness of the rich nutritional value of dairy products is, and the stronger purchase intention for dairy products is. The number in the household has a significantly negative effect on the purchase willingness. The larger the household population is, the weaker willingness to buy dairy products is. This may differ from previous studies, which theoretically suggest that the higher the number of household members is, the higher the demand for dairy

products in the household is, and the corresponding higher the willingness to purchase is. In order to explore the source of this contradiction, we conducted partial telephone callbacks to respondents with a household size of more than five and learned that there is a significant difference in the demand preference for dairy products due to higher household size, which increases the difficulty of purchasing dairy products and for this reason reduces the willingness to purchase. The elderly and children significantly and positively affect the willingness to buy dairy products. When the epidemic occurred, the elderly and children became the key groups to pay attention to. If there are elderly or children in the family, more attention should be paid to the nutritional balance in daily life, so the purchase intention of dairy products will be stronger. Changes in household income significantly and positively affect purchase intention. Income is the basis of consumption, and income level affects consumers' willingness to buy to a certain extent. Therefore, if the income level increases after the outbreak, the willingness to buy dairy products will also be correspondingly promoted. Settlement sites negatively affect purchase intention. That is, compared with cities, rural residents are more willing to buy dairy products after the outbreak. The survey found that due to less going out under the epidemic, the awareness of rural residents has increased significantly, so that it may significantly increase their willingness to buy dairy products. Quality and safety concerns significantly and negatively affect the purchase behavior for dairy products. Dairy product quality and safety incidents will seriously affect purchase behavior. When consumers pay more attention to the quality and safety of dairy products, it is easier to inhibit the increase in the purchase proportion of dairy products. Satisfaction with quality supervision has a significantly positive impact on dairy product purchase intention and purchase behavior. That is, the more satisfied consumers are with the current quality supervision, the stronger purchase intention they will have to buy dairy products; at the same time, the purchase proportion of dairy products will increase. Consumption convenience significantly and positively affects the willingness to purchase dairy products; that is, better consumption convenience contributes to increased purchase intention for dairy products. These variables are consistent with the findings of previous studies and will not be discussed too much here (Table 6).

### *4.3. Analysis of Differences*

According to the above statistical description and empirical tests, it can be concluded that consumer health awareness has a significantly positive effect on dairy product purchase behavior. In order to further analyze the impact of health awareness on dairy product purchase behavior among different groups of consumers, considering an empirical test, this paper divides consumers into groups based on health change perception, health concern degree, and health habit development. The groups with values of "1" and "2" are classified as low health awareness groups, and those with a value of "3" are assigned to the high health awareness group. The Heckman two-stage model is also adopted to estimate the impact of health awareness on the purchase of dairy products among different groups of consumers to test the group differences.

Table 7 shows the test results of group differences based on health change perception. For consumers with low health change perception, gender and education levels significantly and positively affects dairy purchase behavior. For consumers with high health change perception, quality and safety attention significantly and negatively affect dairy product purchase behavior, and quality supervision satisfaction significantly and positively affect dairy product purchase behavior. In general, consumers with low health change perception are more susceptible to the influence of gender and education on purchase behavior of dairy products, and consumers with high health change perception are more likely to be affected by the quality and safety concerns and satisfaction with quality supervision. Therefore, it further illustrates that consumers with different degrees of health change perception after the outbreak of COVID-19 have certain differences in the purchase behavior of dairy products.

**Table 7.** Intergroup Difference Test based on Health Change Perception.

| Variable | Low Health Change Perception | | High Health Change Perception | |
|---|---|---|---|---|
| | Purchase Intention (Probit) | Purchase Behavior (OLS) | Purchase Intention (Probit) | Purchase Behavior (OLS) |
| Gender | 0.1733 * (0.0959) | 0.0247 *** (0.0082) | 0.2452 (0.2093) | 0.0067 (0.0264) |
| Age | −0.0404 (0.0573) | −0.0070 (0.0049) | 0.2911** (0.1301) | −0.0058 (0.0138) |
| Education level | 0.0615 (0.0461) | 0.0158 *** (0.0046) | 0.3626 *** (0.1294) | −0.0148 (0.0222) |
| Family population | −0.1533 *** (0.0526) | −0.0015 (0.0047) | 0.1240 (0.1122) | 0.0009 (0.0156) |
| Changes in household income | 0.4278 *** (0.0927) | −0.0067 (0.0085) | 0.1762 (0.2197) | 0.0339 (0.0282) |
| Is there an old man or a child | 0.1634 *** (0.0500) | −0.0064 (0.0051) | −0.0690 (0.1099) | 0.0048 (0.0153) |
| Settlements | −0.1299 (0.1091) | −0.0021 (0.0096) | −0.5076 ** (0.2405) | −0.0052 (0.0311) |
| Attention to quality and safety | 0.0788 (0.0591) | −0.0028 (0.0063) | −0.0794 (0.1420) | −0.0682 *** (0.0179) |
| Quality Supervision Satisfaction | 0.1656 *** (0.0554) | 0.0021 (0.0060) | 0.3375 ** (0.1474) | 0.0484 *** (0.0202) |
| Consumption convenience | 0.4286 *** (0.0382) | | 0.4365 *** (0.0932) | |
| con. | −1.7740 *** (0.3752) | 0.1317 *** 0.0419 | −3.3134 *** (0.9766) | 0.3725 ** (0.1689) |
| Inverse Mills ratio | −0.0793 *** (0.0200) | | 0.0968 (0.1111) | |
| Wald chi2 | 26.08 | | 20.55 | |
| Pro > chi2 | 0.0020 | | 0.0148 | |
| N | 1306 | | 474 | |

Note: The parentheses are the standard errors corresponding to the estimated coefficients. ***, **, and * indicate significance at the levels of 1%, 5%, and 10%, respectively.

The results of the group differences based on the degree of health concern are shown in Table 8. For consumers with a low health concern degree, the level of education has a significantly positive impact on the purchase behavior for dairy products, and the number in the household has a significantly negative impact on the purchase behavior for dairy products. For the group with high health concern degree, satisfaction with quality supervision has a significantly positive impact on dairy product purchase behavior, while education level and quality and safety concern have significantly negative impacts on dairy product purchase behavior. In general, both the low health concern group and the high health concern group are easily affected by the level of education, but the education level differs in the direction of the impact effect between the low and high health concern groups. At the same time, the dairy product purchase behavior of consumers in the low health concern group is also vulnerable to the influence of the family population, and the high health concern group is also affected by the quality and safety concern and the satisfaction of quality supervision. It can be seen that after the outbreak of the epidemic, consumers with different levels of health concern showed certain differences between groups in their purchase behavior of dairy products.

**Table 8.** Tests for Differences between Groups based on the Degree of Health Concern.

| Variable | Low Health Change Perception | | High Health Change Perception | |
|---|---|---|---|---|
| | Purchase Intention (Probit) | Purchase Behavior (OLS) | Purchase Intention (Probit) | Purchase Behavior (OLS) |
| Gender | −0.0534 (0.1201) | 0.0152 (0.0103) | 0.4530 *** (0.1332) | 0.0180 (0.0149) |
| Age | −0.0603 (0.0680) | −0.0005 (0.0056) | 0.1900 ** (0.0817) | −0.0037 (0.0082) |
| Education level | −0.0092 (0.0542) | 0.0152 *** (0.0048) | 0.2414 *** (0.0761) | −0.0210 ** (0.0115) |
| Family population | −0.1320 ** (0.0651) | −0.0099 * (0.0058) | −0.0285 (0.0706) | 0.0058 (0.0085) |
| Changes in household income | 0.3633 *** (0.1158) | −0.0071 (0.0101) | 0.2658 * (0.1272) | 0.0000 (0.0157) |
| Is there an old man or a child | 0.1196 ** (0.0610) | −0.0013 (0.0061) | 0.1007 (0.0665) | −0.0024 (0.0091) |
| Settlements | −0.1027 (0.1401) | −0.0001 (0.0119) | −0.3829 *** (0.1404) | −0.0081 (0.0174) |
| Attention to quality and safety | −0.0185 (0.0769) | 0.0024 (0.0074) | 0.1122 (0.0772) | −0.0538 *** (0.0110) |
| Quality Supervision Satisfaction | 0.1446 * (0.0741) | 0.0044 (0.0078) | 0.2674 *** (0.0726) | 0.0343 *** (0.0107) |
| Consumption convenience | 0.4049 *** (0.0479) | | 0.4855 *** (0.0530) | |
| con. | −0.9615 ** (0.4551) | 0.0821 * (0.0474) | −3.4753 *** (0.5898) | 0.3855 *** (0.0893) |
| Inverse Mills ratio | −0.054 ** (0.0251) | | −0.0579 (0.0449) | |
| Wald chi2 | 18.34 | | 34.30 | |
| Pro > chi2 | 0.0314 | | 0.0001 | |
| N | 737 | | 1043 | |

Note: The parentheses are the standard errors corresponding to the estimated coefficients. ***, **, and * indicate significance at the levels of 1%, 5%, and 10%, respectively.

Table 9 shows the test results of differences between groups based on the degree of health habit development. For consumers in the low health habit development group, gender and education level significantly and positively affect dairy product purchase behavior. For consumers in the high-health habit formation group, quality and safety attention negatively affects dairy product purchase behavior, and quality supervision satisfaction positively affects dairy product purchase behavior. In general, the dairy product purchase behavior of consumers with low health habit development is easily affected by gender and education level, while the purchase behavior of consumers with high health habit development is more likely to be affected by quality safety concerns and quality supervision. Different satisfaction shows different buying behaviors. Therefore, after the outbreak of the epidemic, consumers with different levels of health habit development also have certain differences between groups in the purchase behavior of dairy products.

**Table 9.** Test of Differences between Groups based on the Degree of Health Habit Formation.

| Variable | Low Health Change Perception | | High Health Change Perception | |
|---|---|---|---|---|
| | Purchase Intention (Probit) | Purchase Behavior (OLS) | Purchase Intention (Probit) | Purchase Behavior (OLS) |
| Gender | 0.2325 ** (0.1140) | 0.0235 ** (0.0105) | −0.0897 (0.1386) | 0.0070 (0.0178) |
| Age | −0.0409 (0.0682) | −0.0036 (0.0065) | 0.0409 (0.0778) | −0.0077 (0.0089) |
| Education level | 0.0984 * (0.0519) | 0.0156 *** (0.0055) | 0.0733 (0.0784) | −0.0182 (0.0127) |
| Family population | −0.1427 ** (0.0592) | −0.0080 (0.0056) | −0.0298 (0.0817) | 0.0139 (0.0109) |
| Changes in household income | 0.4123 *** (0.1043) | 0.0058 (0.0101) | 0.3566 ** (0.1510) | −0.0059 (0.0199) |
| Is there an old man or a child | 0.1304 ** (0.0547) | 0.0008 (0.0061) | 0.0934 (0.0804) | −0.0087 (0.0112) |
| Settlements | −0.1641 (0.1227) | 0.0018 (0.0116) | −0.2113 (0.1659) | 0.0045 (0.0218) |
| Attention to quality and safety | 0.1002 (0.0651) | 0.0048 (0.0074) | −0.1511 (0.1053) | -0.0752 *** (0.0135) |
| Quality Supervision Satisfaction | 0.1303 ** (0.0646) | 0.0036 (0.0073) | 0.3921 *** (0.0898) | 0.0399 *** (0.0147) |
| Consumption convenience | 0.4594 *** (0.0437) | | 0.3997 *** (0.0602) | |
| Con. | −1.9932 *** (0.4358) | 0.0849 (0.0526) | −1.5037 *** (0.5704) | 0.4540 *** (0.0986) |
| Inverse Mills ratio | −0.0773 *** (0.0230) | | −0.1419 ** (0.0653) | |
| Wald chi2 | 14.33 | | 40.85 | |
| Pro > chi2 | 0.1109 | | 0.0000 | |
| N | 1020 | | 760 | |

Note: The parentheses are the standard errors corresponding to the estimated coefficients. ***, **, and * indicate significance at the levels of 1%, 5%, and 10%, respectively.

### 4.4. Robustness Test

In order to further test the robustness of the data, this paper used a combination of principal component analysis and factor analysis to reconstruct the variable of health awareness. The specific method is to combine the health change perception, health concern degree, and health habit development. Principal component factor analysis was performed on the three-dimensional health awareness variables, and the first principal component is extracted and then treated as a common variable again into the Heckman two-stage model to re-examine the impact of consumer health awareness on the purchase behavior of dairy products. By Stata 14.0 statistical analysis software for principal component factor analysis, the variance contribution rate of the extracted first principal component reaches 76.20%, which explains 76.20% of the data information in the original data. On this basis, health awareness is taken as the variable in the model for the second regression, and the results are shown in Table 9. The regression results show that consumer health awareness passes the significance test at the 1% level in both the first and second stages, and the coefficient signs are all positive. These results fully confirm the previous theoretical hypothesis, that is, by further improving consumer health awareness, purchase intention and purchase

behavior can be significantly enhanced under the impact of COVID-19, which also reflects the robustness of the results for the previous regression analysis (Table 10).

**Table 10.** Regression Results of Health Awareness Variables constructed by principal Component Factor Analysis.

| Variable | Purchase Intention (Probit) | Purchase Behavior (OLS) |
|---|---|---|
| Health consciousness | 0.1820 ***<br>(0.0374) | 0.0604 ***<br>(0.0052) |
| Gender | 0.1757 **<br>(0.0863) | 0.0138<br>(0.0098) |
| Age | 0.0060<br>(0.0496) | −0.0056<br>(0.0054) |
| Education level | 0.0758 *<br>(0.0427) | −0.0040<br>(0.0061) |
| Family population | −0.1119 **<br>(0.0472) | −0.0032<br>(0.0057) |
| Changes in household income | 0.3705 ***<br>(0.0841) | 0.0000<br>(0.0102) |
| Is there an old man or a child | 0.1236 ***<br>(0.0444) | −0.0022<br>(0.0060) |
| Settlements | −0.2017 **<br>(0.0971) | −0.0028<br>(0.0115) |
| Attention to quality and safety | 0.0425<br>(0.0532) | −0.0331 ***<br>(0.0073) |
| Quality Supervision Satisfaction | 0.1818 ***<br>(0.0507) | 0.0246 ***<br>(0.0073) |
| Consumption convenience | 0.4142 ***<br>(0.0348) | |
| Con. | −1.6504 ***<br>(0.3412) | 0.2597 ***<br>(0.0523) |
| Inverse Mills ratio | | −0.0440<br>(0.0280) |
| Wald chi2 | | 180.61 |
| Pro > chi2 | | 0.0000 |

Note: The parentheses are the standard errors corresponding to the estimated coefficients. ***, **, and * indicate significance at the levels of 1%, 5%, and 10%, respectively.

## 5. Conclusions

Based on survey data of 1780 consumers in 31 provinces (cities, autonomous regions) in China, this paper uses the Heckman two-stage model to empirically test the impact of consumer health awareness on dairy product purchase behavior under the impact of COVID-19. The differences in the factors affecting consumer dairy product purchase behavior are further analyzed under the condition restrictions of health awareness. The study results show that: (1) 79% of consumers are more willing to buy dairy products after the outbreak, and the purchase proportion of dairy products increases 17.49% on average compared to before the outbreak; (2) Health change perception, health concern degree, and health habit development in consumer health awareness all have an important impact on the purchase behavior of dairy products. Health change perception and health concern degree both positively and significantly affect the purchase intention for dairy products, and health change perception, health concern degree, and health habit development in consumer health awareness have positive effects on the purchase proportion increase of dairy products; (3) The difference analysis shows that there are obvious differences

among consumer groups with different health awareness in the purchase behavior of dairy products; (4) Through principal component factor analysis, it was found that the whole consumer health awareness directly has an important impact on the purchase intention and the purchase proportion increase for dairy products.

In order to enhance purchase intention and purchase behavior for dairy products from the perspective of health awareness and to quickly restore the kinetic energy of dairy product consumption, this paper proposes the following policy recommendations based on the above conclusions: First of all, it is necessary to enhance consumer awareness of dairy product nutrition and improve the purchase intention and the purchase proportion of dairy products. According to the statistical results, although most consumers are more willing to buy dairy products after the outbreak, there is still an improvement space. Government departments should positively guide consumers for the awareness of nutrition and health of dairy products through media promotion and brand promotion of marketing companies. In this way, the purchase intention and the purchase proportion can be improved subjectively. Secondly, it is necessary to carry out dynamic tracking of consumer health awareness and adjust dairy product supply strategies in a timely manner. The empirical results show that health change perception, health concern degree, and health habit development all have important impacts on the purchase behavior of dairy products. As the epidemic enters the normalization stage of prevention and control, there are still sporadic cases in many parts of the country, and health awareness will also continue to change. In the post-epidemic period, dairy companies should continue to pay attention to the changes in consumer health awareness, conduct follow-up investigations, and adjust dairy product supply strategies based on opportunities, links, and changes to meet the consumption demand of different consumers; Third, identify different health-awareness consumer groups and implement differentiated marketing strategies for dairy products. The empirical results based on health change perception, health concern degree, and health habit development show that there are significant differences in factors affecting the purchase behavior of dairy products under limited conditions. Therefore, consumer groups with low health awareness should improve cultural cognition, while high health awareness consumer groups should strengthen their satisfaction with quality and health.

There are some shortcomings in this paper. We did not pay special attention to the impact of consumer health awareness on dairy product purchase behavior before the outbreak of COVID-19, so we cannot accurately compare the changes in consumer health awareness and dairy product purchase behavior before and after the outbreak of COVID-19. However, we try our best to ensure that the changes in consumer health awareness are caused by the impact of COVID-19. The questionnaire was designed with the prominent words "what I would do after the outbreak of COVID-19" to emphasize that both the change in consumer health awareness and the change in dairy product purchase behavior were caused by the outbreak of COVID-19. In the future, we will continue to focus on this issue and explore the impact of changes in consumer health awareness on dairy product purchase behavior in stages. This paper belongs to a study conducted on the initial period of the post-epidemic period. We will continue to follow up with these 1780 consumers in the middle and later periods of the post-epidemic period to observe the changes in their health awareness and dairy product purchase behavior. It is hoped that these conclusions can contribute a modest contribution to restoring the momentum of dairy consumption, thus improving dietary nutrition and ensuring the health of the people.

**Author Contributions:** Conceptualization, J.X.; Methodology, J.X.; Software, J.W.; Validation, J.X.; Formal Analysis, J.X. and J.W.; Resources, J.W.; Data Curation, J.W.; Writing—Original Draft Preparation, J.X.; Writing—Review and Editing, J.X. and C.L.; Funding Acquisition, C.L. All authors have read and agreed to the published version of the manuscript.

**Funding:** This work was funded by the National Natural Science Foundation of China (The effect of infant milk powder safety trust index on product competitiveness—Index measurement, Correlation model construction and market simulation, Project number 71673042).

**Institutional Review Board Statement:** Not Applicable.

**Informed Consent Statement:** Informed consent was obtained from all subjects involved in the study.

**Data Availability Statement:** The datasets analyzed during the current study are not publicly available due to the data being from a survey of 1780 consumers in China but are available from the corresponding author on reasonable request.

**Conflicts of Interest:** The authors declare no conflict of interest.

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
