# Peer review of "Impact of Consumer Health Awareness on Dairy Product Purchase Behavior during the COVID-19 Pandemic"

_sustainability, doi:10.3390/su14010314_

Round 1

Reviewer 1 Report

I would like to express my gratitude to the authors and the Editor for providing me with the opportunity to review this manuscript. I carefully read it and believe it has the potential to be published in this journal, but only after substantial revisions. The research topic is intriguing, and it makes use of a large consumer dataset for analysis. However, in almost every section of the manuscript there is still a lot of room for improvement. The following are my comments and suggestions:

General Comment

  • The article does a poor job in terms of clarity, standard terminology, language fluency, and grammar usage. Throughout the manuscript, there are many incomplete sentences, inconsistent use of terms, and unclear points. Here are a few examples:

Line 17: “The percent of 79% of consumers preferred to….”

Line 26: “ ….found that, in overall, consumer health awareness….”

Line 73-77: “……. purchase intention based on both expectation and perception are significant in predicting purchase behavior [11]. purchase intention based on both expectation and 76 perception are significant in predicting purchase behavior.”

Line 78-79: “An individual health 78 awareness with healthy lifestyle [12].”

Line 140: “Health awareness to four dimensions, namely,….”

  • The above are only few examples of incorrect, incomplete, or duplicated sentences. There are many more to be found throughout the manuscript. Please see following examples for non-standard usage of terminology:

Table 1: Statistical categories (should be Variables)

Table 1: Class indication (should be Level)

Table: Family population (should be Family size)

Line 349: “Identify variables…” should be Instrumental Variable (IV).

Line 410: “….. has passed the significance test at the level of 1%...”

The same incorrect terminology is used throughout the manuscript.

  • Given the preceding examples, I would advise the authors to have a professional English editor go over the manuscript and standardize the use of terminologies.

Introduction

The introduction is a little too long while also lacking in clarity. The authors should make it shorter and conclude with a statement of research objectives, followed by the research gap. Also, please move some of the content from the discussion section (where the authors describe the study's innovations) to the introduction section.

Materials and Methods

  • One intriguing aspect of this study is that it makes use of a large dataset gathered from across the country. The specifics of the data collection, on the other hand, remain completely unclear. For example, how were the questionnaires administered, who collected the data, when the data was collected (month-year), how long it took to collect the data, whether it was collected online or through personal interviews, what sampling strategy was used, and how many respondents were chosen from each province or municipality. These additions will help to improve the methods sections. I would also suggest that the authors include a table describing the proportion of sample size chosen from each province/municipality.
  • The timeliness of data collection is the second major concern in this regard. The authors use the phrase "after the outbreak" several times. So, what constitutes "after"? What was the cut point considered? How long did the buffer period before data collection last? This is an important question to answer because there are still sporadic outbreaks in many parts of China, and some areas have been placed under lockdown at various times since early 2020.
  • Some variables are used to quantify change (as compared to before the outbreak). So, how were the questions posed to establish causality, that is, whether this change was truly caused by the epidemic? This aspect is mentioned by the authors in one place in the manuscript, but it should be included in this section as well.
  • Finally, the choice of instrumental variable (referred to by the authors as the identity variable) must be justified, as the currently chosen variable (convenience of consumption) appears to be affection both intention and purchase behavior. Please elaborate on the selection of this variable for the Heckman two-stage model.

Analysis of Empirical Results

  • First of all, please rename this section "Results and Discussion" and then create sub-sections under that heading.
  • In all cases, the description of model results requires more explanation. In Table 5, for example, age has a negative relationship with purchase behavior, but the presence of elderly at home may improve purchase intention (as per results). Because behavior is driven by intention, the two outcomes are appear to be contradictory. Please provide more information.
  • Why does having a large family have a negative relationship with purchase intention? The authors do not explain this, instead simply stating that the relationship is negative. This needs to be looked into further.
  • What does the "settlements" variable in several Tables mean? It would be appropriate to include units or levels with the variables as needed.
  • Gender has a positive and significant relationship with the dependent variable in an intergroup difference test based on health change perception, degree of health concern, and health habit formation (Table 6, 7, 8). Could you please explain what that means?
  • What were the items extracted from the principal component analysis to form the variable of health awareness? In Table 9, the authors refer to this variable as "health consciousness," but in the text, it is referred to as "health awareness." Please elaborate.

Discussion

The discussion section does not include any discussion, only the study's innovations. I would suggest that the authors combine some of these points in the Introduction section, rename the previous section "Results and Discussion," and discuss the results alongside the discussion following the Tables with empirical results.

Conclusions

  • The authors state that data was collected from 30 provinces, but it is previously stated that data was collected from 31 provinces. Please correct this statement.
  • Please include the study's limitations in this section.
  • Please use the phrase "after the outbreak" with caution because you have not collected panel data on two different time periods. Perhaps you can use "consumers' self-reported increase in consumption following the outbreak.”

Reviewer 2 Report

Dear author(s),

Thank you for the invaluable read.

Following are some of the suggestions:

There are many claims that should be properly cited. For instance, Line 34-38. At the beginning of 2020, COVID-19 Epidemic (COVID-19) broke out in Wuhan, 34
China, and then gradually spread to all parts of the country. Although the domestic epidemic has been basically controlled, the epidemic still has a long-term impact on the dairy product market. In the early stage of the epidemic, when the national economic development pressing the “pause button”, resident consumption levels and consumer confidence continued to decline. 
Again Line 46-47: Where is the source for such claim? From January to September 2020, the total profit of dairy product processing was 22.359 billion yuan, a year-on-year decrease of 13.63%.
There should be proper justification for preferring dairy product industry.
Provide justification why 1780 questionnaire were selected. Above 200 would have been enough to draw justification. 
Is there any specificity about why there are more females in comparison to males? Do you think female consumers are more affected? Why not equal distribution of sample population.
How did you deal with common method bias?
What were the steps taken to ensure the validity and reliability. 
There are formatting errors, kindly ensure you proofread before submission. For instance, Line 555 and 562. Point 2 and Point 3. After digit there should be a space. 
Findings and discussion section should be incorporated with the literature at hand. The results should always be discussed in the light of literature so you accept or contradict the past research findings.
There should be future directions for the researchers. 

Round 2

Reviewer 1 Report

The authors have made major revisions in a satisfactory manner in response to my comments. Therefore, I recommend that this paper should be accepted for publication.